# Long-Term Outcome in Systemic Lupus Erythematosus; Knowledge from Population-Based Cohorts

**DOI:** 10.3390/jcm10194306

**Published:** 2021-09-22

**Authors:** Sigrid Reppe Moe, Hilde Haukeland, Øyvind Molberg, Karoline Lerang

**Affiliations:** 1Department of Rheumatology, Oslo University Hospital, 0424 Oslo, Norway; hihauk@ous-hf.no (H.H.); oyvind.molberg@medisin.uio.no (Ø.M.); klerang@ous-hf.no (K.L.); 2Department of Rheumatology, Martina Hansens Hospital, 1346 Gjettum, Norway; 3Institute of Clinical Medicine, University of Oslo, 0424 Oslo, Norway

**Keywords:** epidemiology, Systemic Lupus Erythematosus, outcome, mortality, survival, end-stage renal disease, cancer

## Abstract

Background: Accurate knowledge of outcomes in Systemic Lupus Erythematosus (SLE) is crucial to understanding the true burden of the disease. The main objective of this systematic review was to gather all population-based studies on mortality, end-stage renal disease (ESRD) and cancer in SLE. Method: We performed a systematic literature search in two electronic databases (MEDLINE and Embase) to identify all population-based articles on SLE and survival, mortality, ESRD and cancer. The SLE diagnosis had to be verified. We used the Preferred Reporting Items for Systematic Reviews and Meta-Analyses guidelines (PRISMA). Results: We included 40/1041 articles on mortality (27), ESRD (11) and cancer (3), of which six were defined as inception studies. In the total SLE cohort, the standardized mortality ratio ranged from 1.9 to 4.6. Cardiovascular disease was the most frequent cause of death in studies with follow-up times over 15 years. SLE progressed to ESRD in 5–11% of all SLE patients. There are no data supporting increased cancer incidence from population-based inception cohorts. Conclusion: There is a need for more population-based studies on outcomes of SLE, especially inception studies, with the use of control groups and follow-up times over 15 years.

## 1. Introduction

Systemic Lupus Erythematosus (SLE) is a rare systemic and chronic disease often referred to as the prototype of autoimmune rheumatic diseases because of the varied spectrum of clinical manifestations and diversity of phenotypes. The etiology of SLE is believed to be multifactorial, and both genetic predisposition and environmental triggers are most likely involved [1]. The incidence, severity and phenotypic expression of the disease differ between ethnic groups, gender and age at disease onset. The annual incidence of SLE varies from 0.3 to 23.3/100,000, and the prevalence varies from 0 to 241/100,000 [1]. The variations are highly dependent on the method of retrieval and the definition of SLE diagnosis.

Several aspects of SLE make it one of the most challenging conditions to study at the population level. First, no diagnostic criteria for SLE exist and the diagnosis is based on the judgement of an experienced clinician. Diagnosing SLE can be challenging since SLE is a great imitator of other diseases. The symptoms of SLE overlap many other diseases that can easily be mistaken for SLE in as much as 40% [2,3,4,5] of cases.

Secondly, in many countries, SLE patients are not treated in the same hospital and/or specialization since different organs may be affected and the severity of the disease varies. Selected patient populations from tertiary hospitals tend to miss milder cases, and therefore underestimate the incidence and overestimate the severity of SLE. Thus, a closer estimate of the true frequency of clinical and laboratory SLE manifestations and outcomes is more likely from a geographically complete cohort of patients. All these aspects of the disease make it difficult and labour-intensive to collect epidemiological data. In Georgia, Lim et al. found 45,000 potential patients, screened 3142 records and found 1320 patients with a verified SLE diagnosis. In Sweden, Ingvarsson et al. screened 2461 cases and found 55 patients with a verified diagnosis, and Voss et al. in Denmark screened 980 cases to find 95 patients with a verified SLE diagnosis [4,5,6].

Earlier publications on SLE and epidemiology differ greatly in study-design. A good epidemiological study is highly dependent on valid data to obtain reliable results that are indicative of the total size of the problem and thus, a reliable assessment of outcome. Truly population-based research, with a verified and ascertained SLE diagnosis by chart review, is the best way to achieve the most accurate knowledge possible on this disease and its outcome measures. The use of standardized methods gives the best basis for comparison of epidemiological data across different studies and countries.

The objective of this study was to conduct a review of literature on population-based epidemiologic data on SLE and well-defined and hard outcomes; mortality, end-stage renal disease (ESRD) and cancer. The elected publications were thoroughly reviewed to ensure that they were from population-based cohorts and that the SLE diagnose was verified.

## 2. Materials and Methods

A senior medical librarian searched two electronic databases: MEDLINE (Ovid) and Embase (Ovid), from their inception to 25 June 2021, with language restricted to English. The systematic search used both controlled vocabulary (MeSH terms or EMTREE terms) and text word search in title, abstract or author keywords. The search consisted of two searches with different approaches. Search 1: Concepts for systemic lupus, SLE criteria, mortality or cancer, were combined with the Boolean operator AND. Search 2: Concepts for lupus nephritis, end stage renal disease or kidney transplantation were combined with the Boolean operator AND (Appendix A). Both searches were restricted to population-based cohorts.

Two investigators (HH and KL or SRM and KL) independently evaluated all abstracts and titles to determine eligibility for inclusion. When necessary, the articles were reviewed in full, and, if in conflict, discussed in plenum (HH, SRM, KL). The authors also searched the reference list of included articles to find additional relevant studies.

For inclusion in this systematic review, the SLE diagnosis had to be verified by chart review. Studies on SLE were included on the relevant outcomes: mortality, overall and renal survival and risk of malignancy.

We excluded: (1) Studies that failed to validate the SLE diagnosis by chart review; (2) Studies based on administrative data; (3) Studies from tertiary centers only, if it was not specified that it was the only hospital serving the region; (4) Animal studies; (5) Meta-analysis; (6) Case reports; (7) Studies on unrelated outcomes; (8) Studies of selected SLE subsets (paediatric SLE, biopsy-proven lupus nephritis (LN), hospital inpatients); (9) Studies with fewer than 30 patients; (10) Studies on subset of relevant outcome (cardiovascular mortality).

Causes of death analyses were excluded from this review if the study reported only multiple causes of death. We defined the study period as years from start of inclusion to end of follow-up. The total SLE population was defined as all SLE patients in the given study-period. Incident SLE were defined as patients diagnosed within the study-period. Inception SLE was defined as patients diagnosed within the study-period and captured within one year of the diagnosis.

This review was carried out in accordance with the Preferred Reporting Items for Systematic Reviews and Meta-Analyses (PRISMA) guidelines [7].

## 3. Results

We screened 1041 titles/abstracts. Through the screening process, we identified 40 studies that met the criteria for inclusion, whereof 27 were for survival and mortality, 11 were for ESRD and three were for cancer (Figure 1). We found seven articles through manual search of the reference list of included articles. The case finding methodology and SLE ascertainment in all included cohorts is described in Appendix A. All but three study locations included only SLE patients who fulfilled four or more of the American College of Rheumatology SLE classification criteria [8,9,10,11].

### 3.1. Standardized Mortality Rate and Survival

Twenty-three population-based studies reported survival with SLE, while a standardized mortality rate (SMR) was reported in 13 studies. Eighteen studies used incident patients for survival analysis, while five included all SLE patients (total). Six studies used only incident patients and seven used the total SLE population for SMR analysis (Table 1).

The ten-year survival in incident cohorts ranges from 46% in Curacao to 92% in northern Norway, and from 90 to 92% in Europe and 76 to 89% in North America [10,12,13,14,15,16,17,18,19,20,21,22,23,24,25,26,27]. Five and ten-year survival differed in incident cohorts with patient inclusion before and after 1990 (five-year survival 80% versus 92% and ten-year survival 63% versus 88%) [10,12,13,14,15,16,17,18,19,20,21,22,23,24,25,26,27,28]. For all studies with patient inclusion starting after 1990, the five-year survival was 90% or more, except for Barbados and Wisconsin [9,11,14,20,23,27,29]. In studies on total SLE cohorts, the SMR ranges from 1.9 to 4.6 [9,19,20,26,27,30,31,32]. For female SLE patients, the SMR ranges from 1.8 to 4.7, while in male patients the SMR ranges from 1.5 to 4.6 [9,10,19,20,30,31,33]. There was no significant difference between the two groups. Among the incident SLE patients the SMR varied from 1.3 to 11.1, depending on follow-up time (one to 33 years) [10,16,17,33,34]. Only one incident study reported 25-year survival with SLE (60% survival versus 73% in the general population) [10].

### 3.2. The Main Causes of Death in Systemic Lupus Erythematosus

An average of 41% of patients in the studies from Asia died of infections, compared to an average of 12% in studies from Europe (Table 2) [9,10,14,16,17,20,21,23,30,33,35]. Renal failure was the underlying cause of death in about 17% (median) of SLE patients, except for a much higher frequency in Barbados (46%) [12,16,21,23,25,27,30,33,35]. From the article with the shortest follow-up time versus the longest, the causes of death varied from 60% infections and 6% cardio- and cerebrovascular disease (CVD) in Hong Kong [33] to 15% infections and 59% CVD in Sweden [10]. CVD was the most frequent cause of death in the two study locations with population-based cohorts over time [9,10].

### 3.3. End Stage Renal Disase

Within the primary studies reviewed, ESRD developed in 5–11% of the total SLE patients [35,36,37], of which 5–6% were in a Scandinavian population (Table 3) [36,38]. The incidence rate of ESRD varied from 2.3 to 11.1/1000 patient years in incident patient populations, depending on the population studied (Table 3) [38,39,40].

### 3.4. Cancer

We found only three studies on cancer in population-based cohorts, from three different countries. Only the study from Sweden was an inception study (Table 4) [41].

## 4. Discussion

The literature search on outcomes in SLE and mortality, ESRD and cancer revealed population-based studies from 22 different locations around the world. The main discovery is that from 1990 there is a higher survival rate during the first five to ten years of the disease. A cardiovascular cause of death is common later in the disease’s course, and improvement in survival is less clear. Death caused by infections differs between geographical area and the death rate due to infections is lowest in Europe. Development of ESRD occurs in 5–10% of SLE patients in cohorts of European and Asian ethnic population. ESRD is, however, more common in the African ethnic population. We only discovered one study on cancer from a population-based cohort with inclusion at the time of the SLE diagnosis.

It is well established that the change in treatment of SLE after the 1950s and 1960s caused the survival rate to improve tremendously, from less than a 50% survival rate over five years in the 1950s [47,48]. There are, however, some aspects of selected patient populations that may influence the reported outcome; a tertiary center may overestimate the severity of SLE by missing the diagnosis of milder SLE cases, for example.

Our search on survival with SLE revealed a ten-year survival rate varying with time and location, from 46% in Curacao in the 1980s to 93% in a more recent study from northern Norway [15,19]. The overall trend in survival indicates an improvement in five- and ten-year survival rates after 1990, with a five-year survival similar to the control population. This discovery is in accordance with the conclusion in a recent meta-analysis that survival with SLE improved up to the 1990s, but since appears to have stabilized [48].

A control group is necessary to enhance the quality of survival estimates in SLE. As survival from SLE improves, it may become similar to the survival rate in the general population. The reported survival rate from studies depends on the age composition of the SLE cohort and hence, the time since inception. In this systematic review, nine of the studies included made use of a control group in their survival analysis. They all included only incident cases and five studies were also defined as inception studies. From the inception studies with control groups conducted after 1990, the ten-year survival is only slightly lower in the SLE groups versus the control groups (91% vs. 96%) [19,20]. However, the gap seems to increase with time from diagnosis [10,21].

Findings from this review also indicate that the main causes of death from SLE differ with the length of follow-up time of the studies; CVD is more frequent in studies with the longest follow-up time [10,21]. It is well known from earlier studies that death due to CVD is more frequent later in the course of the disease [49,50]. Urowitch et al. identified this bimodal pattern of mortality in 1976 [51]. In the included studies, European SLE patients died less often of infections compared to Asians. It appears we still do not manage to prevent CVD over time, as up to 59% of SLE patients die of CVD. This might indicate better treatment for the acute phase of SLE, but not for damage accrual due to SLE. However, death from infections remains prominent in certain parts of the world.

In this review, SMR in total SLE cohorts ranges from 1.9 to 4.6, similar, but with a slightly lower range of variation, compared to two previous meta-analyses [52,53]. Studying SMR in incident populations makes comparison difficult as the inclusion periods differ, the highest SMR being from Taiwan within the first year after diagnosis [34]. Several studies have identified ethnicity as a modifier of outcome in SLE, with lower survival in patients of African descent [47,54]. This corresponds with our findings of the lowest SMR in a predominantly white Scandinavian population. The discrepancy in prognosis might be due to both genetic and socioeconomic factors. A possible gender disparity in SLE prognosis has been proposed; however, the results have been inconsistent and contradictory [55]. In this review, we found no significant sex differences in SMR.

Many studies have reported the risk of ESRD development in SLE, and, as registries of biopsy-proven LN are quite common, outcomes in this particular patient subset have been widely investigated. However, as many as 44% of all LN patients are not biopsy-proven [56]. Thus, we excluded studies of this selected SLE patient subset, as they might differ from other LN patients. In this review, we found that only 11 population-based studies estimated the frequency of ESRD in SLE populations. An estimated 5–11% of SLE patients progressed to ESRD, fewer than in a recent meta-analysis [57]. A lower frequency of ESRD in the white population is in line with previous reports [54,57]. The trend in ESRD development seems to be stable over time, despite improvements in therapy. This corresponds to findings from a recent meta-analysis where the risk of ESRD development remained unchanged during the last decade [57].

We identified only three studies on cancer development in SLE patients. Only one was an inception study [41]. In these studies, the cancer risk was increased by 1.2–1.8 times. By comparison, a prior review, which also included non-population-based studies, found an increased risk of cancer ranging from 1.1 to 3.6 times in the SLE population [58]. The lowest cancer risk (SMR 1.2) found in our review was from an old Swedish study with 116 SLE patients. The study from the National Health Insurance Research Database from Taiwan is on the other end of the scale, with a SIR of 1.8 [46].

Earlier studies, mostly non-population-based or without a verified diagnosis, have found that hematological cancers appear more often in the SLE population compared to the general population [58]. All three studies in this review found significantly higher numbers of lymphomas, and especially non-Hodgkin lymphomas, with a reported SMR of 11.6 from Sweden and SIR of 7.3 from Taiwan [41,45,46]. In addition, all three studies found an increased incidence of lung cancer [41,45,46]. Taiwan reports a significant increase for lung/mediastinum (SIR 1.2) [46], yet data from Sweden (SMR 5.6) and Iceland (O/E ratio 1.7) are not significant [41,45].

Cancer development in SLE patients is particularly difficult to study for two reasons. First, cancer sometimes leads to death; subsequently, patients who get cancer early in the course of the disease may not be captured. Secondly, some people with cancer might have paraneoplastic symptoms that may mimic SLE and then be mistakenly diagnosed with SLE. This emphasizes both the importance of a verified SLE diagnosis in studies on cancer and SLE, and the need for further population-based, and preferably inception-based (early capture), studies on cancer.

Considerable differences in the methods for case finding, verification of diagnosis, and study design can make comparing the results of the SLE outcomes difficult. To overcome some of these problems, all studies in this systematic review have employed comprehensive case-finding and case ascertainment methods, or it has been indicated in the article that all patients in a defined geographic region were included. However, the geographic area and its location for care of SLE patients is not always described in detail, and it is likely that we have missed some population-based studies.

The composition of the cohorts used for analysis of outcomes differs as some studies include all patients and some include only incident patients, making comparisons more difficult. Only seven studies of incident SLE patients had a follow-up period over 15 years [10,13,14,21,25,27,41]. The reason for this may be that hospital data registries going back before the year 2000 are rare and not so easily accessible. They may also not contain the entire volume of ICD-codes on outpatients [59].

Most of the population-based studies, except for Taiwan, are small due to the work effort necessary to identify all patients and verify their diagnoses. Taiwan has a good health system, and 96%–99% of its population is included in the National Health Insurance Database. All SLE patients must fulfill the ACR criteria to receive their benefit claim checks as in- and outpatients [34,39,42,43,46]. However, this may also give the patients and their doctors an additional motive towards approving the SLE diagnosis. In addition, verification of the SLE diagnosis is processed earlier on in the course of the disease in Taiwan compared to the other studies. Hence, an early misdiagnosis of SLE would not be reclassified retrospectively.

We found that six locations (Iceland, Lund in Sweden, Funen in Denmark, northern Norway, Rochester in the USA and New Territories in Hong Kong) have repeated the retrieval of patients at several time points [10,19,27,31,33]. Scandinavia is highly represented in publishing from population-based studies, probably due to the health care system being mostly public, making it easier to identify the patients. Despite small study populations, these are valuable contributions to population-based knowledge of outcomes for SLE. Lund in Sweden already published the very first data on survival from a population-based cohort in 1989 and has, to date, the longest follow-up time on an inception cohort reporting on 25-year survival (60%) [10]. However, four locations from the USA have made a tremendous effort collecting larger population-based cohorts that were published in the last decade [25,27,32,37,40].

## 5. Conclusions

Population-based studies on SLE patients with a verified diagnosis is considered the gold standard in the pursuit of finding the true outcomes of suffering from SLE. Studies using the 1997 ACR criteria are easier to compare over time, as most studies included only SLE patients with four or more ACR criteria. There is a special need for cancer studies and studies with longer follow-up time on survival in population-based inception cohorts.

## Figures and Tables

**Figure 1 jcm-10-04306-f001:**
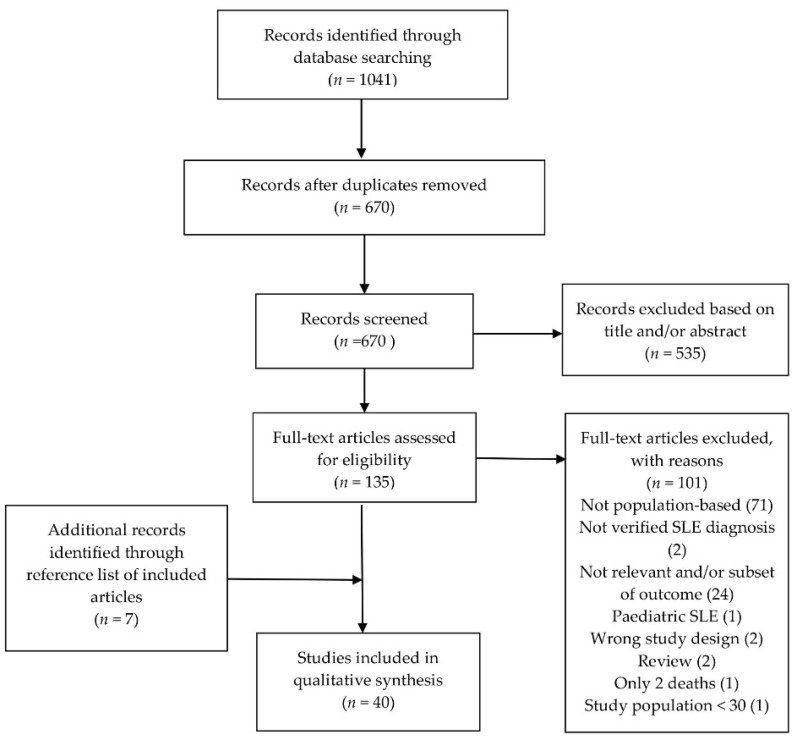
Flowchart of literature search and study inclusion. Studies identified through MEDLINE (Ovid) and Embase (Ovid) through 25 June 2021.

**Table 1 jcm-10-04306-t001:** Survival and standardized mortality rate (SMR) in Systemic Lupus Erythematosus, from population-based cohorts.

Author; Year (Ref.)	Study Location	Study Period *	Ethnicity	Follow-Up Time	SLE Cases, *n*		Deaths, *n*		SMR, 95% CI			Survival (Controls)%			
					Total	Incident	Total	Incident	Total	Female	Male	5 Years	10 Years	15 Years	20 Years
	**NORTH AMERICA**														
Peschken et al.; 2000 [13]	Canada	1980–1997	177 Caucasian	NA	257	257	NA	NA				97	95	91	
			49 NAI									95	83	75	
Uramoto et al.; 1999 [26]	MN, USA	1950–1979	Mainly Caucasian	7.2 years (µ)	79	79	NA	NA	2.7 (1.7–4.2)			75 (95)	50 (92)		
		1980–1992										90 (90)	71 (90)		
Naleway et al.; 2005 [24]	WI, USA	1991–2001	Mainly Caucasian	5.8 years (µ)	117	44	NA	8				88	76		
Bartels et al.; 2014 [25]	WI, USA	1991–2009	NA	7.7 years (µ)	70	70						87 (90)	74 (81)	59 (73)	
Jarukitsopa, S et al.; 2015 [27]	MN, USA	1993–2005	80% white	7.8 years (µ)	117	45	NA	6	2.6 (1.0–5.6)			93	89	64	
Lim et al.; 2019 [32]	GA, USA	2002–2016	76% black	NA	1689	336	401	97	3.1 (2.8–3.4)	3.1 (2.8–3.5)	3.0 (2.3–3.9)				
Flower et al.; 2012 [12]	Barbados	2000–2009	98% African Caribbean	NA	183	183	24	24				88	80		
	**SOUTH AMERICA**														
Lucero et al.; 2020 [29]	Argentina	2005–2012	83% Mestizos	NA	353	NA	32	NA				96	93		
Nossent; 1992 [15]	Curaçao	1980–1990	All of African descent	NA	94	68	25	NA				60	46		
	**ASIA**														
Iseki et al.; 1994 [35]	Japan	1972–1993	NA	4877 PY	566	NA	104	NA				89	78	72	69
Mok et al.; 2005 [14]	Hong Kong, China	1991–2003	All ethnic Chinese	NA	258	258	29	29				92	83	80	
Mok et al.; 2008 [33]	Hong Kong, China	2000–2006	Mainly Asian	NA	442	NA	30	NA	3.9 **	4.0 **	9.6 **				
Yeh et al.; 2013 [34]	Taiwan	2003–2008	NA	NA	6675	6675	1611	1611	11.1 (NA)						
Al-Adhoubi et al.; 2021 [11]	Oman	2006–2020	NA	NA	1160	NA	54	NA				100	100		99
	**EUROPE**														
Gudmundsson et al.; 1990 [17]	Iceland	1975–1988	NA	NA	76	76	17	17	3.4 (2.0–5.4)			84	78		
Jacobsen et al., 1999 [30]	Denmark	1975–1995	NA	4185 PY	513	NA	122	NA	4.6 (3.8–5.5)	4.7 (3.9–5.8)	4.0 (3.8–5.5)	91	76		53
Nossent et al.; 2001 [18]	Norway, north	1978–1999	>96% Caucasian	NA	105	83	18	11				92	75		
Eilertsen et al.; 2009 [19]	Norway, north	1978–1995	98.8% Caucasian	NA	81	81	25	25	2.0 (1.4–2.8)	2.1 (1.5–3.1)	1.5 (0.6–3.5)	91(98)	81 (96)		
		1996–2007	98.3% Caucasian	NA	58	58	5	5				96 (98)	92 (96)		
Lerang et al.; 2014 [20]	Norway, Oslo	1999–2009	84% of European descent	2665/812 PY	325	129	50	7	3.0 (2–3.8)	2,7 (2,0–3.7)	4.6 (2.3–8.1)	95 (99)	90 (96)		
Jonsson et al.; 1989 [28]	Sweden, Lund	1981–1986	NS	342 PY	86	38	9	NA				97 (97)			
Ståhl-Hallengren et al.; 2000 [22]	Sweden, Lund	1981–1991	NA	NA	162	162	17	17				93 (98)	83 (96)		
Ingvarsson et al.; 2019 [10]	Sweden, Lund	1981–2014	98.3% Caucasian	3053 PY	174	174	60	60	2.5 (1.9–3.3)	2.7 (2.0–3.6)	1.9 (1.0–3.4)	91 (97)	85 (91)	73 (86)	62 (77)
Alamanos et al.; 2003 [16]	Greece	1982–2001	NA	NA	178	178	12	12	1.3 (NA)			97	90		
Alonso et al.; 2011 [21]	Spain	1987–2006	NA	7.8 years (µ)	150	150	19	19				94 (97)	87 (94)	80 (89)	
Laustrup et al.; 2009 [31]	Denmark, Funen	1995–2003	94% Caucasian	767 PY	138	NA	15	NA	1.9 (1.0–3.0)	1.8 (0.9–3.2)	2.1 (0.4–6.2)				
Voss et al.; 2013 [9]	Denmark, Funen	1995–2010	94% Caucasian	2052 PY	215	NA	38	NA	2.3 (1.6–3.2)	1.9 (1.3–2.9)	3.2 (1.5–6.3)	94	73		
Pamuk et al.; 2015 [23]	Turkey	2003–2014	NA	48 months (mdn)	331	331	17	17				95	90		

SLE: Systemic Lupus Erythematosus, SMR: Standardized Mortality Rate, PY: Patient years, µ: Mean, mdn: Median, CI: Confidence Interval, NA: Not avaliable, GA: Georgia, WI: Wisconsin, MN: Minnesota, NAI: Native American Indians. * Years from start of inclusion to end of follow-up, ** Calculated. Inception SLE: Incident SLE cases captured within one year from diagnosis. Total SLE: All SLE cases within the given study-period. Incident SLE: SLE cases diagnosed within the study-period.

**Table 2 jcm-10-04306-t002:** The main causes of death in Systemic Lupus Erythematosus, from population-based studies.

Author; Year (Ref.)	Study Location	Study Period *	Follow-Up Time	Deaths/SLE Cases; *n*/*N*	Cause of Death, %					
					Active SLE	CVD	Infections	PD	Malignancy	Renal Failure
	**NORTH** **AMERICA**									
Bartels et al.; 2014 [25]	WI, USA	1991–2009	540 patient years	19/70		32%	16%		13%	13%
Jarukitsopa et al.; 2015 [27]	MN, USA	1993–2005	7.8 years (mean)	6/45			67%			33%
Flower et al.; 2012 [12]	Barbados	2000–2009	NA	24/181			42% ^d^	8%		46% ^b^
	**SOUTH** **AMERICA**									
Lucero et al.; 2020 [29]	Argentina	2005–2012	NA	32/353			44%			
	**ASIA**									
Mok et al.; 2005 [14]	China	1991–2003	NA	29/258		31% ^c^	55%		3%	
Iseki et al.; 1994 [35]	Japan	1972–1993	4877 patient years	104/566		15%	24%			12%
Mok et al.; 2008 [33]	China	2000–2006	NA	30/422		6%	60%	3%		7%
	**EUROPE**									
Jacobsen et al.; 1999 [30]	Denmark	1975–1995	4185 patient years	122/513	19%	24%	20%		7%	10%
Voss et al.; 2013 ^a^ [9]	Denmark	1995–2010	2052 patient years	38/214	8%	32%	8%	16%	13%	
Gudmundsson et al.; 1990 [17]	Iceland	1975–1988	NA	17/76	35% ^b^	29%	6%			
Ingvarsson et al.; 2019 ^a^ [10]	Sweden	1981–2014	3053 patient years	60/174	7%	59%	15%	5%	13%	
Alamanos et al.; 2003 [16]	Greece	1982–2001	NA	12/178			17%			17%
Alonso et al.; 2011 [21]	Spain	1987–2006	7.8 years (mean)	19/150		21%	21%		26%	11%
Lerang et al.; 2014 [20]	Norway	1999–2009	2665 patient years	50/325	12%	16%	6%		20%	
Pamuk et al.; 2015 [23]	Turkey	2003–2014	48 months (mdn)	17/331		24%	23% ^d^		12%	12%

CVD: Cardio- and cerebrovascular Disease, PD: Pulmonary Disease, mdn: Median, NA: Not available, WI: Wisconsin, MN: Minnesota. * Years from start of inclusion to end of follow-up ^a^ Last articles of multiple over time, ^b^ Death attributed to Lupus Nephritis, ^c^ Including hemorrhagic stroke, ^d^ Sepsis. Total SLE: All SLE cases within the given study-period. Incident SLE: SLE cases diagnosed within the study-period.

**Table 3 jcm-10-04306-t003:** Risk of End Stage Renal Disease in Systemic Lupus Erythematosus, from population-based studies.

Author, Year (Ref.)	Study Location	Study Period *	Follow-Up Time	SLE Cases, *n*		LN,%	Age, Years	Ethnicity	ESRD Development		
				Total	Incident				Total SLE	Incident SLE	LN
	**NORTH AMERICA**										
Somers et al., 2014 [37]	MI, USA	2002–2004	NA	2129	399	32	All	56% black patients	Total 10.8%; black 15.3%, white 4.5%		
Plantinga et al., 2016 [40]	GA, USA	2002–2004	2603 patient years	344	344	NA	All	76.1% black patients		Total 11.1; black 13.8, white 3.3/1000 patient years	
	**ASIA**										
Iseki et al., 1994 [35]	Japan	1972–1991	4788 patient years	566	NA	49	All	NA	9%		
Yu et al., 2016 [39]	Taiwan	2000–2008	NA	1196	1196	NA	All	NA		6.1/1000 patient years	
Lin et al., 2017 [42]	Taiwan	2000–2011	8.1 years (mean)	7326	7326	NA	All	NA		4.3%	
Lin et al., 2013 [43]	Taiwan	2003–2008	NA	4130	4130	NA	All	NA		2.5%	
	**EUROPE**										
Jacobsen et al., 1998 [36]	Denmark	1975–1995	8.2 years (mean)	513	NA	42	All	NA	5%		
Eilertsen et al., 2011 [44]	Norway	1978–1995	NA	62	62	32	≥16	98% Caucasian		10 years renal survival: 100%	
		1996–2007	NA	87	87	18	≥16	99% Caucasian		10 years renal survival 88.5%	
Jonsson et al., 1989 [28]	Sweden	1981–1986	NA	86	38	30	≥15	NA			3.8%
Gergianaki et al., 2017 [3]	Greece	1999–2013	7.2 years (mean)	750	NA	13	≥15	97% Greek			4.4%
Reppe Moe et al., 2019 [38]	Norway	1999–2017	18.4/10.6 years (mean)	325	129	30	≥16	84% of European descent	6%	2,3/1000 patient years	

SLE: Systemic Lupus Erythematosus, LN: Lupus Nephritis, ESRD: End Stage Renal Disease, GA: Georgia, MI: Minnesota, NA: Not available. * years from start of inclusion to end of follow-up. Inception SLE: Incident SLE cases captured within one year from diagnosis. Total SLE: All SLE cases within the given study-period. Incident SLE: SLE cases diagnosed within the study-period.

**Table 4 jcm-10-04306-t004:** Cancer risk in Systemic Lupus Erythematosus relative to the general population, from population-based studies.

Author; Year (Ref.)	Study Location	Study Period *	Follow-Up Time, Mean	Age, Years	SLE Cases, *n*		SLE Cases with Malignancies, *n*	Malignancies O/E Ratio (95% CI)	Subgroups of Malignancy **, O/E Ratio (95% CI)/(*p*) ***
					Total	Incident			
	**EUROPE**								
Ragnarsson et al.; 2003 [45]	Iceland	1957–2001	12.8 years	All	238	NA	27	O/E 1.4 (0.9–1.9)	Skin SCC 6.4 (1.3–18.5) Lymphoma 5.5 (0.6–19.6) Lung 1.7 (0.4–5.0) Breast 1.6 (0.7–3.2) Prostate 1.2 (0.0–6.2)
Nived et al.; 2001 [41]	Sweden	1981–1998	9.4 years	>15	116	116	11	SMR 1.2 **** Male 2.2 (0.6–5.7) Female 1.0 (0.4–2.1)	NHL 11.6 (1.4–42) Prostate 6.4 (1.3–18.7) Lung 5.6 (0.7–20.1)
	**ASIA**								
Chen et al.; 2010 [46] ^a^	Taiwan	1996–2007	6.1 years	All	11,763	11,763	259	SIR 1.8 (1.7–1.8)	NHL 7.3 (7.0–7.6) Vagina/vulva 4.8 (4.2–5.3) Nasopharynx, siunus, ears 4.2 (3.9–4.5) Leukemia 2.6 (2.5–2.8) Skin 1.7 (1.6–1.8) Breast 1.6 (1.5–1.6) Cervix 1.4 (1.3–1.5) Lung/mediastinum 1.2 (1.2–1.3) Prostate 0.8 (0.7–0.9)

SLE: Systemic Lupus Erythematosus, O/E ratio: observed/expected events, SMR: Standardized Morbidity Rate, SIR: Standardized Incidence Ratio, RR: Relative Risk, NA: Not available, NHL: Non-Hodgkin Lymphoma, SCC: Squamous Cell Carcinoma. * Years from start of inclusion to end of follow-up, ** Not all results included, *** *p* < 0.001 indicates statistical significance, **** Calculated. ^a^ Main article on cohort, sub-analysis not included. Inception SLE: Incident SLE cases captured within one year from diagnosis. Total SLE: All SLE cases within the given study-period. Incident SLE: SLE cases diagnosed within the study-period.

## Data Availability

Not applicable.

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
