# Peer review of "Long-Term Outcome in Systemic Lupus Erythematosus; Knowledge from Population-Based Cohorts"

_jcm, 2021, doi:10.3390/jcm10194306_

Round 1

Reviewer 1 Report

1. This is interesting review paper, but I consider what was inappropriate in abstracts/titles that 535 studies were excluded? This is difficult to verify some papers based only on title or abstract due to word restrictions in many journals.

2. Secondly, most of analyzed studies included patients with SLE recognized based on ACR classification criteria, that were established in 1982 and slightly revised later in 1997. Authors of this manuscript included however the study from Oman which enrolled patients who met SLICC criteria. I do not think that such paper should be included to the review. What is more,  studies from Lund in Sweden included patients with clinical diagnosis of SLE and I recommend to omit those Swedish studies.

3. Thirdly, why pediatric studies were included, since the course of the disease in children is different?

Reviewer 2 Report

This manuscript consolidates results of population-based studies on long term outcomes in mortality, ESRD and cancer. 

Comments:

  1. Results: Tables 1,2 & 3 would be easier to read if the studies are ordered according to study location in continents/regions, then chronologically according to earliest date of study period e.g. Table 1: Uramoto [23] & other USA studies, followed by European studies Gudmundsson [13] etc.
  2. Discussion L187-88: Antibiotic resistance as the reason for Asian SLE patients dying from infection more often than European SLE patients seems unlikely. Asian SLE patients in general have more severe disease than European patients and require more immunosuppression, with the associated complications of severe infection. Did you compare these regions according to the same decades? Do these studies provide some evidence about the difference in SLE manifestations and treatment between Europe and Asia?
  3. Grammatical errors to be corrected: Materials/methods L89 - defined the study period; Results: L128 - 5-6% were in; Table 4 L149 - statistical significance; Discussion L185 - Urowitz; L191 - death from infections; L231 - paraneoplastic

Reviewer 3 Report

this is a systematic review including the population-based studies on SLE, whose objective was to assess the survival, mortality, ESRD, and cancer in this disease. It is an important topic.

Introduction: "First, no diagnostic criteria for SLE exist and the diagnosis is based on the judgment of an experienced clinician" (line 36). Probably, the authors wanted to say that there are not diagnostic criteria, but classification criteria, however, a few phrases below, they write: "Lim et al in Georgia found 45 000 potential patients, screened 3142 records and found 1320 patients fulfilling the ACR SLE criteria. Ingvarsson et al in Sweden screened 2461 cases and found 55  fulfilling the criteria, and Voss et al in Denmark screened 980 cases to find 95 patients fulfilling the criteria" (lines 47-50) and, lately, in Methods section (line 78-79), "SLE population was to be determined by chart review and SLE diagnosis weighed by validated classification", therefore these criteria were used for diagnosis.

"Thus, hospital diagnosis (i.e. register studies) of SLE is likely overestimating the prevalence of SLE with as much as 40%" - interesting assertion, as we know that the criteria were changed in order to be more sensitive (SLICC), because SLE diagnosis was missed. Indeed, the latest criteria from 2019 became very specific, too. 

How was the search restricted to population-based cohorts?

Please explain in the Methods section and below the tables what exactly means Total SLE and incident SLE / Total cases, Incident cases, and Inception cases.

Discussion

"The main discovery is that from 1990 there is a higher survival rate the first five-ten years of the disease". This is not demonstrated in the Results section.

A condition of validity for prognostic studies is the inception cohort, therefore it is not clear how useful and reliable are the results from total SLE cases, and if they should have been included.

It is known that SLE has two mortality peaks, the first due to infections and SLE complications, and the second due to cardiovascular disease. Therefore, one cannot compare (or even assess) mortalities from long with short follow-up, the inception cohorts with short follow-up will not have cardiovascular mortalities, while the results of the cohorts of total cases are uninterpretable. However, I find that the Discussions section addresses well these facts.
